# The Combination of MoS_2_/WO_3_ and Its Adsorption Properties of Methylene Blue at Low Temperatures

**DOI:** 10.3390/molecules25010002

**Published:** 2019-12-18

**Authors:** Yifan Zheng, Jingjing Wang, Yedan Wang, Huan Zhou, Zhiying Pu, Qian Yang, Wanzhen Huang

**Affiliations:** 1College of Chemical Engineering, Zhejiang University of Technology, Hangzhou 310014, China; zhengyifan@zjut.edu.cn (Y.Z.); wang17857@163.com (J.W.); yifanyibufan_wyd@163.com (Y.W.); huanzhou@zjut.edu.cn (H.Z.); zhiyingpu@zjut.edu.cn (Z.P.); yangqian@zjut.edu.cn (Q.Y.); 2Research Center of Analysis and Measurement, Zhejiang University of Technology, Hangzhou 310014, China

**Keywords:** MoS_2_/WO_3_, combination, adsorption, methylene blue

## Abstract

It was found previously that neither monomer MoS_2_ nor WO_3_ is an ideal material for the adsorption of organic dyes, while MoS_2_/WO_3_ composites synthesized by a two-step hydrothermal method have outstanding adsorption effects. In this work, the chemical state of each element was found to be changed after combination by X-ray photoelectron spectroscopy analysis, which lead to their differences in adsorption performance. Moreover, the adsorption test of methylene blue on MoS_2_/WO_3_ composites was carried out under a series of temperatures, showing that the prepared composites also had appreciable adsorption rates at lower temperatures. The adsorption process could be well described by the Freundlich isothermal model and the pseudo-second order model. In addition, the particle-internal diffusion model simulation revealed that the internal diffusion of the particles played an important role in the whole adsorption process.

## 1. Introduction

With the development of some chemical industries, such as textile, leather, paper, plastics, more and more organic dyes are used in the manufacturing process in order to color the products and also consume a substantial volume of water. As a result, a lot of organic dye wastewater has been produced [1]. According to relevant statistics, more than 100,000 dyes have been produced each year, and the total production exceeds 7 × 10^5^ tons. Even if the content of organic dyes in the water is less than 1/1,000,000, it can cause uneasiness [2]. Methylene blue (MB) is an azo dye often used in printing and dyeing textiles. It is also frequently used as a pH control for chemical reactions in the laboratory, industry, and agriculture. MB is a micro-toxic type, so a large amount of oral administration can cause abdominal discomfort, skin eczema, and have irritating effects on the eyes. Therefore, it is meaningful to treat wastewater containing MB [3]. At present, there are many physical, chemical, and biological methods used in the treatment of organic dye wastewater, such as flocculation precipitation [4], hydrogen peroxide treatment [5], and photo electrocatalytic degradation [6,7]. Chemical adsorption is a method of adsorbing nonionic organic refractory compounds in dye wastewater [8]. Because of its simple operation and low cost, more and more attention has been paid to the chemical adsorption in dye wastewater. The development of efficient adsorbents is the key to improving adsorption efficiency. Nowadays, the commonly used adsorbents in the adsorption field are carbon, minerals, chitosan, and other organic compounds.

MoS_2_ is a new graphene-like material. Its layered structure is sandwiched together by Mo–S–Mo units with a weak van der Waals force [9], and the direct band gap is 1.82 eV [10]. This unique structure gives MoS_2_ stable chemical properties, an adjustable band gap, a large specific surface area, and an abundant layered edge to provide a large number of unsaturated active sites [11]. Therefore, MoS_2_ is expected to become a high-performance adsorbent. In addition, by controlling the preparation parameters and methods, the morphology and number of layers of the MoS_2_ can be adjusted, and heterojunctions can be constructed to expose more active edge sites for adsorption. Sabarinathan et al. used the hydrothermal method to synthesize layered MoS_2_ with citric acid as an auxiliary additive [12]. It was found that an appropriate amount of citric acid induced the growth of the original MoS_2_ microspheres into a well dispersed layered structure. With the appearance of such a dispersed layered structure, the photocatalytic performance of MoS_2_ for methylene blue was more excellent. Hou et al. prepared MoS_2_/WO_3_ composite semiconductor photocatalysts successfully by the hydrothermal method and studied the photocatalytic performance on MB [13]. After compounding with tungsten oxide, the photocatalytic effect of MoS_2_ on MB was better than that of pure tungsten oxide and molybdenum disulfide. Unfortunately, the reaction still took a long time to complete and there were certain restrictions, such as the use of visible light. Moreover, the studies only focused on transferring photo-induced electrons but ignored capturing the holes. Therefore, it is essential to introduce a reasonable bandgap-matchable semiconductor material as a template to construct an MoS_2_-coated system, which can provide holes for trapping electrons [14,15,16,17,18,19], facilitating charge transfer and thus enhancing adsorption ability.

In our previous work, the MoS_2_/WO_3_ heterojunction was successfully synthesized by a two-step hydrothermal method using rod-like tungsten oxide as a template, and the adsorption performance under dark conditions at room temperature was proven to be excellent. For further investigation, in this work, X-ray photoelectron spectroscopy (XPS) was used to study the combination of MoS_2_ and WO_3_. Considering the practical situation of application, the adsorption performance of the composites under a series of temperatures was also investigated. Furthermore, the adsorption mechanism was also discussed by the adsorption thermodynamics and kinetic simulation.

## 2. Results and Discussion

### 2.1. The Phase of Samples

The crystal phases of the obtained samples were investigated by X-ray diffraction (XRD) analysis. The corresponding XRD patterns of WO_3_, MoS_2_, and MoS_2_/WO_3_ are presented in Figure 1. It can be seen that all diffraction peaks of pattern c were well indexed to WO_3_ (Joint Committee on Powder Diffraction Standards, JCPDS No. 75-2187) and MoS_2_ (JCPDS No. 17-0744), which preliminarily indicates that the MoS_2_/WO_3_ composites were successfully produced.

### 2.2. The Morphology of Samples

Figure 2a shows the SEM images of WO_3_, which exhibited a uniform rod-like structure randomly oriented with a length of a few microns or more. As shown in Figure 2c, after reacting for 24 h, there were interlaced MoS_2_ sheets dispersed on the WO_3_ nanorods, expanding the diameter of the nanorods. Moreover, a porous, discontinuous, and loose surface could be found in MoS_2_/WO_3_ (Figure 2c). It is this unique structure that leads to the excellent adsorption capacity of MoS_2_/WO_3_ composites, which has already been discussed by Ying et al. [20]. Besides this, the SEM image of MoS_2_ prepared without adding WO_3_ as a template under the same conditions is shown in Figure 2b, which indicates that the product exhibited microspheres formed by sheet-like subunits. Simultaneously, the element mapping of as-prepared MoS_2_/WO_3_ sample in Figure 3 showed a uniform distribution of Mo and S on the surface of W and O, which further proves that the coating composites were formed.

### 2.3. The Elemental Chemical States Analysis of Samples

As is known, the adsorption reaction is closely related to the surface state of the sample and the interface of the catalysts. Thus, it is significant to investigate the surface states of obtained samples by XPS measurements. The XPS results are presented in Figure 4. As for Mo_3d_ in MoS_2_, shown in Figure 4A_1_, two obvious peaks appeared at 229.2 eV and 232.3 eV, which were attributed to Mo 3d_5/2_ and Mo 3d_3/2_, and the pair of peaks at higher binding energies (229.6 eV and 233.2 eV) demonstrates that there is no-metric ratio MoS_x_ in mono MoS_2_. Comparably, two peaks of Mo 3d_5/2_ and Mo 3d_3/2_ in MoS_2_/WO_3_ (Figure 4C_1_) were slightly shifted to higher binding energies (about 0.3 eV). This suggests that there are strong electronic interactions between MoS_2_ and WO_3_ domains through established heterogeneous interfaces [21,22]. In addition, the pair of peaks in Figure 4C_1_ at about 233.5 eV and 236.4 eV, corresponding to Mo 3d_5/2_ and Mo 3d_3/2_, suggests the presence of Mo^6+^. Comparing O 1s in WO_3_ before and after combination, the peaks at 527.4 eV (Figure 4B_1_) and 531.1 eV (Figure 4C_3_) indicate the formation of W–O binding. After combination, a new peak was found in 531.9 eV, which means there was a new chemical bond generated. This peak can be attributed to O^2−^ in the Mo–O bond, which is in agreement with the results of Mo 3d in MoS_2_/WO_3_. Cui et al. proved that with the increase in MoO_3_ composition in the transformation of MoS_2_ to MoO_3_, the product showed better adsorption capacity [23]. Besides, the peaks at the binding energy of 529.0 eV in Figure 4B_1_ and 532.6 eV in Figure 4C_3_ were both due to the adsorbed O on the surface of the samples. In Figure 4C_4_, the peaks at about 36.3 eV and 35.2 eV of W 4f_7/2_ suggest the formation of W–O and W–S bindings. S 2p spectra in the obtained samples (Figure 4A_2_ and Figure 4C_2_) could be deconvoluted into three pairs of peaks (S 2p_3/2_, S 2p_1/2_), corresponding to MoS_2_, MoS_x_, and W–S–Mo respectively. The S 2p_3/2_ peak was attributed to the typical metal-S bond, while the S 2p_1/2_ peak was corresponding to the sulfur with low coordination, which is generally related to sulfur defects [24,25]. W–S and Mo–S have a similar structure, so it is usually assumed that tungsten catalysts are similar to molybdenum catalysts. At the edge of this layered structure, the sulfur vacancies, which are active sites for catalytic reactions, are easily created because of the weaker sulfur bonding [26]. The semi-quantitative data of Mo–O and W–S were obtained by XPS data fitting results. Within the range of measurement on the surface of the sample, the Mo–O accounted for 10% of the total and the W–S accounted for 5%. Consequently, we conjectured that the formation of the W–S–Mo and Mo–O–W structures promotes the MoS_2_ layers being more tightly coated on WO_3_ nanorods, which is beneficial for increasing the specific surface area in favor of adsorption.

In short, the peaks of Mo3d, S2p, O1s, and W4f all shifted to higher binding energies after combination, indicating that there are strong electronic interactions between MoS_2_ and WO_3_. There were new bonds of Mo–O and W–S generated, suggesting the chemical combination of MoS_2_ and WO_3_. Therefore, this results in the high stability of MoS_2_/WO_3_ composites. Moreover, as discussed above, the S 2p_1/2_ peak revealed the presence of terminal unsaturated S atoms on the Mo–S and W–S sites. Consequently, associated with SEM, TEM, and XPS analysis, the as-synthesized MoS_2_/WO_3_ composites were proved to possess a porous nanostructure and defected interface of rich active sites and are expected to achieve excellent adsorption performances.

The details of this unique structure affecting its adsorption property has been discussed in our previous work [20]. In addition, the Zeta potentials of MoS_2_/WO_3_ were measured and are shown in Appendix A, which demonstrates that the adsorbents dispersed in water must have large negative Zeta potentials (−29.3 mV) to have an electrostatic interaction with MB, which has positive charges located on the heteroatom ring. This is another reason why the MoS_2_/WO_3_ composites have an impressive adsorption property on MB.

### 2.4. Effect of Temperature on Adsorption Properties

The effect of temperature on the adsorption rate is revealed in Figure 5. At all experimental temperatures, the MB adsorption rate was higher at the beginning and gradually decreased as contact time increased. This may be attributed to the change in concentration, which is the driving force of adsorption reaction, and the change in available adsorption sites [27]. At the beginning of the reaction, a higher concentration of MB caused large mass transfer and the large difference between the concentration of the volume solution and the adsorbent could produce a large adsorption force, which would promote the transport and adsorption of MB molecules onto the active sites of MoS_2_/WO_3_ [28]. When the temperature of the adsorption reaction decreased from 25 °C to 0 °C, the adsorption capacity within 62 min was studied. Significantly, it only takes about 60 min for MB to be adsorbed on MoS_2_/WO_3_ with the degradation rate up to 98%, which is more efficient than polypropylene–clinoptilolite composites [29], graphene [30], and some other carbon materials [31]. The equilibrium adsorption capacity slightly decreased from 98.6 mg/g to 83.7 mg/g as reaction temperature decreased. These results indicate that the main factor affecting the adsorption rate was not the ambient temperature, but the active center of the material itself. There are a large number of adsorption activity sufficient points on MoS_2_/WO_3_, which determines that it still has favorable adsorption efficiency at lower temperatures. Figure 6 indicates the effect of reaction temperature on the MB degradation rate. The degradation rate remained steady when the reaction temperature varied from 25 °C to 0 °C. However, as observed in Figure 6, the degradation efficiency became lower with the decrease in temperature. According to the theory of adsorption kinetics, the molecular diffusion rate increases with the increase in temperature, and the elevated temperature can accelerate the reaction rate. Therefore, as the temperature decreased, the adsorption efficiency was slightly deteriorated. It is worth noting that the adsorption capacity of the composite could reach 83.7 mg/g in 62 min, even at 0 °C, indicating that MoS_2_/WO_3_ may be an ideal adsorbent for MB and other dyeing materials because of its faster adsorption rate at low temperatures.

### 2.5. Adsorption Kinetics

In order to study the steps that may limit the adsorption rate, the experimental data at different temperature conditions were fitted using the pseudo-first order kinetic equations, the pseudo-second order kinetic equations, and the intra-particle diffusion model. The correlation coefficient (R^2^) was used to express the degree of fitting. Besides this, the corresponding rate constant and theoretical equilibrium adsorption amount were calculated.

The pseudo-first order kinetic model, the pseudo-second order kinetic model, and the intra-particle diffusion model are represented by the following equations (Equation (1), Equation (2), Equation (3)):(1)$${\ln(q}_{e}-q_{t}{)=lnq}_{e}-k_{1}t,$$
where k_1_ indicates the rate constant of the pseudo-first order kinetic (min^−1^);
(2)$$\frac{t}{q_{t}}=\frac{1}{k_{1}q_{e}^{2}}\;+\;\frac{t}{q_{e}},$$
where k_2_ indicates the rate constant of the pseudo-second order kinetic (g·mg^−1^·min^−1^);
(3)$$q_{t}{=k}_{p}t^{1/2}{\;+\;C}_{i},$$
where k_p_ indicates the rate constant of the intra-particle diffusion model (g·g^−1^·min^−1^), while C_i_ is a constant reflecting the thickness of the boundary layer and a larger C_i_ indicates a greater boundary layer effect [32].

The linear fits of the different kinetic models are shown in Figure 7 and the specific parameters are listed in Table 1. For all temperatures, the linear regression correlation coefficient (R^2^) for the pseudo-second order was much higher than that of the pseudo-first order and the intra-particle diffusion model, indicating better fitness while using the pseudo second-order model. This means that extreme membrane diffusion may not be the rate limiting step for MB adsorption on MoS_2_/WO_3_ [33,34]. In addition, the calculated values of q_e_ by the pseudo-second order model were analogous to the corresponding experimental values. Therefore, the pseudo-second order model is more appropriate to describe the MB adsorption onto MoS_2_/WO_3_ and it contains all processes in which MB adsorbs on MoS_2_/WO_3_, such as external liquid membrane diffusion, internal particle diffusion, and the adsorption of adsorbate on the interior surface site of adsorbents [35]. As shown in Equation (3), the intra-particle diffusion model is usually used to further study the possible adsorption mechanism on adsorbents [36]. The total rate of adsorption may be controlled by one of the steps or a combination of more steps [37]. If the linear fitting image of q_e_ versus t^1/2^ is a straight line passing the origin, it means that the adsorption process is a single diffusion. However, multi-linear plots are observed in Figure 7a and none of them passed through the origin. This indicates that the intra-particle diffusion is not the sole rate-limiting step and multiple rate-control steps may be involved in the adsorption process of MB onto MoS_2_/WO_3_. Consequently, we speculate that the adsorption process of MB onto MoS_2_/WO_3_ under different temperature conditions can be divided into three stages. The initial stage is generally related to the external mass delivery, called external mass transfer, which diffuses or moves to the outer surface of the adsorbents through the boundary layer around the adsorbed particles [38,39]. In the next two stages, the first is the diffusion of the adsorbate in the pores and adheres to the surface of the adsorbents, followed by adsorption/desorption equilibrium. In the last stage, the adsorbed organic molecules may exchange or share electrons with the adsorbed particles, depending on the chemical nature of the solid. The relevant parameters are listed in Table 1. According to the values of the intra-particle diffusion rate constants (k_p_) and C_i_ values, intra-particle diffusion plays an important role in the overall adsorption of all groups.

### 2.6. Adsorption Thermodynamics

The adsorption isotherm of the composite material of methylene blue was studied under the condition of 25 °C, 10 mg MoS_2_/WO_3_, and the adsorption time of 24 h. The variation range of the initial concentration of MB solution was from 2 to 40 mg/L (2 mg/L, 5 mg/L, 10 mg/L, 15 mg/L, 20 mg/L, 25 mg/L, 30 mg/L, 35 mg/L, 40 mg/L). The most common expressions describing the solid–liquid adsorption isotherm are the Langmuir equation, the Freundlich equation, and the Tenkin equation. The validity of the isotherm models fitting can be determined by their linearization curve, as shown in Figure 8. The regression coefficient (R^2^) was used as a criterion to compare the fitness of each model.

The linear form of Langmuir isotherm equation is expressed as the following (Equation (4)):(4)$$\frac{C_{e}}{q_{e}}=\frac{1}{{kq}_{m}}\;+\;\frac{C_{e}}{q_{m}},$$
where C_e_ (mg/L) and q_e_ (mg/L) are the equilibrium concentration of the adsorbate in liquid-phase and solid-phase, respectively; Q_m_ (mg/g) is the maximum monolayer coverage capacity adsorbent; and k (L/mg) is the Langmuir adsorption constant. One of the most important characteristics of the Langmuir adsorption isotherm equation is that the dimensionless separation factor R_L_ is defined, which is expressed as in Equation (5):(5)$$R_{L}\;=\;\frac{1}{{1\;+\;kC}_{0}},$$
where R_L_ is used to represent the properties of adsorption process. If 0 < R_L_ < 1, it indicates that the adsorption process is preferential adsorption.

The linear expression of Freundlich isotherm equation is shown in Equation (6):(6)$${\ln q}_{e}={lnK}_{i}\;+\;\frac{1}{m}{lnC}_{e},$$
where K_i_ (mg/L) is a constant to describe the sorption capacity, and m is a constant representing the favorability of the sorption system, and the value of 1/m indicates the effect of concentration on the adsorption capacity [40,41].

The Temkin is one of the isotherm fitting equations that similar to the Freundlich isotherm. The model equation is expressed as the following (Equation (7)):(7)$$q_{e}=Aln(B\cdot C_{e}),$$
where A is a constant related to heat of adsorption, B (L/g) is the Temkin isotherm equilibrium binding constant.

The constants obtained by fitting the three isotherms are summarized in Table 2. According to the regression coefficients, the Freundlich isotherm model was the best to describe the adsorption data, which implies that multi-layer adsorption with uneven adsorption energy may occur in the MB–MoS_2_/WO_3_ system. In this work, the value of 1/m was between 0.1 and 0.5 (1/m = 0.2052), which indicates favorable adsorption of MB onto MoS_2_/WO_3_ [42]. Moreover, a uniform distribution of bounding energy rather than a uniform adsorption energy may also have taken place in the MB–MoS_2_/WO_3_ system, because the regression coefficients for the Temkin isotherm model were relatively high (R^2^ = 0.9779). As shown in Figure 8, when the concentration of MB at equilibrium was within the range of 2~15 mg/L, the Langmuir isotherm model also fitted well. This phenomenon may be due to the Langmuir isotherm being valid for the low ion concentrations and the monolayer adsorption capacity of adsorbents is insufficient. The comparison of maximum adsorption capacities of MB on some other reported adsorbents are shown in Table 3. It can be seen that MoS_2_/WO_3_ prepared in this work exhibited higher adsorption capacity, which is expected to be an ideal adsorbent in a worse environment.

## 3. Materials and Methods

### 3.1. Fabrication of WO_3_ Nanorods

The preparation process of rod-like WO_3_ can be found in Reference 20 for details [20].

### 3.2. Synthesis of MoS_2_/WO_3_ Composites

The formation process of MoS_2_/WO_3_ composites was similar to that of Ying et al. [20]. Typically, 90 mg sodium molybdate (Na_2_MoO_4_·2H_2_O) and 180 mg thioacetamide (C_2_H_5_NS) were dissolved in 60 mL deionized water to form a transparent solution. While stirring, 60 mg WO_3_ nanorods was added into the above solution and then the pH was adjusted to 1.5 with a certain concentration of hydrochloric acid. Then, a suspension was obtained and transferred into a 100 mL Teflon-lined autoclave and then maintained at 220 °C for 24 h. The product was centrifuged and washed three times with deionized water and anhydrous ethanol, and then dried at 60 °C for 20 h.

### 3.3. Characterization

The crystallization and the phase composition were characterized by X-ray diffraction (XRD) with a PAN analytical X’ Pert PRO X-ray diffractometer (PANanalytical B.V., Almelo, The Netherlands) using Cu Kα radiation (λ = 1.5418 Å). The X-ray tube voltage and current were set at 40 kV and 40 mA with a step size of 0.033°/s. The morphology and element mappings of MoS_2_/WO_3_ composites were determined with Hitachi S-4800 scanning electron microscopy (SEM, Tokyo, Japan) and a Tecnai G2 F30 S-Twin transmission electron microscopy (TEM, FEI Company, Hillsborough, OR, USA), respectively. The chemical state of each element of the composite material was characterized by X-ray photoelectron spectroscopy (XPS, Kratos Axis Ultra DLD, Shimadzu Kratos, Manchester, UK). For XPS characterization, spectra representing the C 1s, O 1s, S 2p, Mo 3d, and W 4f photoelectrons were recorded. The test results were processed by XPSPEAK4.1 software (4.1, Hong Kong, China). All binding energies (BE) were calibrated based on the C_1s_ of adventitious carbon (284.8 eV). A UV-visible spectrophotometer (Shanghai Jinghua 7600, Shanghai, China) was adopted to detect the adsorption spectrum of the sample.

### 3.4. Adsorption Test

The adsorption experiments were conducted under dark conditions at a range of different temperatures to test the adsorption process of organic dye onto MoS_2_/WO_3_. Typically, MoS_2_/WO_3_ adsorbent was added to 100 mL MB solution, and then the mixture was magnetically stirred. After a certain time, about 5 mL of suspension was taken out from the stirred mixture and filtered to remove the adsorbent. Then, the resulting solution was measured by a UV-vis 7600 spectrophotometer (Shanghai Jinghua Instruments) at the wavelength range of 400~800 nm. For the efficient study of adsorption, 10 mg MoS_2_/WO_3_ was dispersed in a 100 mL MB solution of 10 mg/L. The mixture was continually stirred at different temperature (from 25 °C to 0 °C) for 62 min and then the absorbance was measured in order to compare the effect of temperature on adsorption rate. Furthermore, different initial concentrations of MB solution were stirred continuously at 25 °C for 24 h to ensure that the adsorption/desorption equilibrium was reached, and then the absorbance was measured to study the thermodynamic properties of adsorption. The degradation of MB was quantified from the decrease in the intensity of the associated characteristic absorption band at 664 nm. The quantity of MB adsorbed on MoS_2_/WO_3_ within a certain period time was calculated by mass balance, as shown in Equation (8), and the degradation rate of MB was determined using Equation (9):(8)$$q_{t}=\frac{C_{0}{-C}_{t}}{m}\;\times \;V,$$
(9)$$\eta =\frac{C_{0}{-C}_{t}}{C_{0}}=\frac{A_{0}{-A}_{t}}{A_{0}},$$
where q_t_ (mg/g) is the pollutant concentration in solid phrase at time t (min), m is the weight of adsorbent (g), V (mL) is the volume of MB solution, and η is the degradation rate, C_0_ (mg/L) and C_t_ (mg/L) are the concentrations of MB at time 0 and t (min), A_0_ is the initial adsorbance of the solution, A_t_ is the adsorbance of each time period.

## 4. Conclusions

In general, MoS_2_/WO_3_ composites prepared by the two-step hydrothermal method with rod-shaped WO_3_ as the template have good adsorption efficiency and are expected to be used as an ideal adsorbent to remove organic dye from wastewater at low temperatures, as well as at room-temperature. According to SEM and TEM images, the composites of MoS_2_-coated WO_3_ nanorods were smoothly obtained. The surface of the structure was loose, which can provide a large number of active sites for the adsorption reaction. The XPS results showed that besides MoS_2_ and WO_3_ in the products, there were also new chemical bonds formed. The chemical structures Mo–S–W and W–O–Mo, newly generated, both had good electro-active sites, so the ability of the composite material to adsorb positively charged dye was greatly improved. It can be concluded from the adsorption thermodynamics and kinetic simulation results that the adsorption of MB onto MoS_2_/WO_3_ is easy to carry out. Moreover, the adsorption capacity of the adsorbent and adsorption efficiency were found to be mildly affected by temperature, even at 0 °C. All the above results show that the as-prepared MoS_2_/WO_3_ composites would have broad prospects of application for removing organic dyes from wastewater.

## Figures and Tables

**Figure 1 molecules-25-00002-f001:**
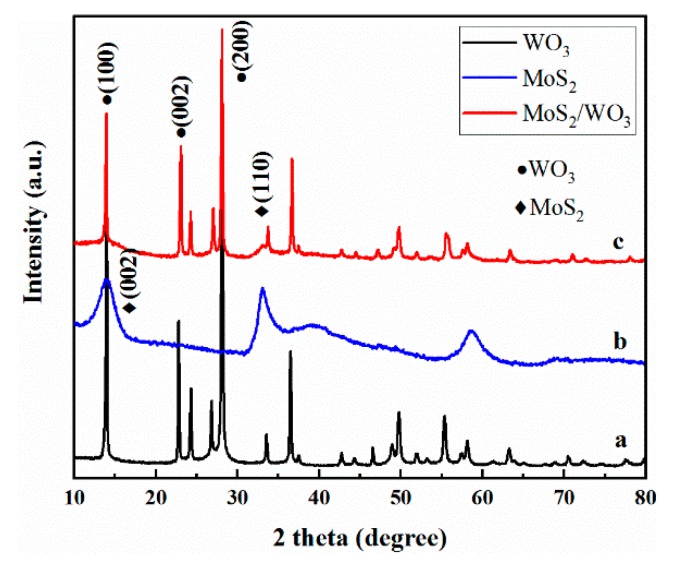
X-ray diffraction (XRD) patterns of WO_3_ (**a**), MoS_2_ (**b**), and MoS_2_/WO_3_ (**c**).

**Figure 2 molecules-25-00002-f002:**
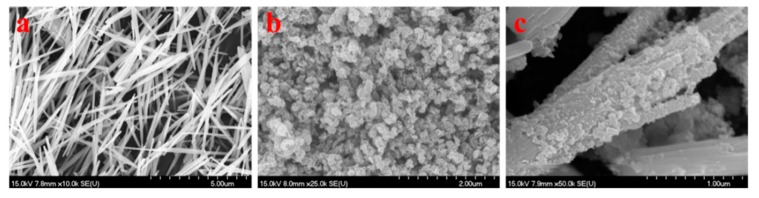
Scanning electron microscopy (SEM) images of WO_3_ (**a**), MoS_2_ (**b**), and MoS_2_/WO_3_ (**c**).

**Figure 3 molecules-25-00002-f003:**
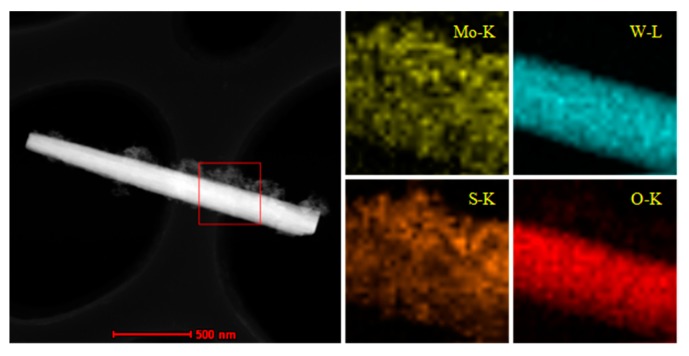
Element mapping of MoS_2_/WO_3_.

**Figure 4 molecules-25-00002-f004:**
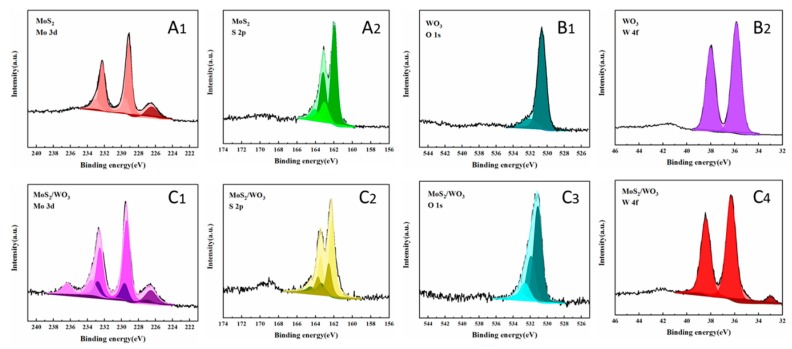
X-ray photoelectron spectroscopy (XPS) spectra of Mo 3d, S 2p, O 1s, and W 4f in the as-prepared MoS_2_, WO_3_, and MoS_2_/WO_3_.

**Figure 5 molecules-25-00002-f005:**
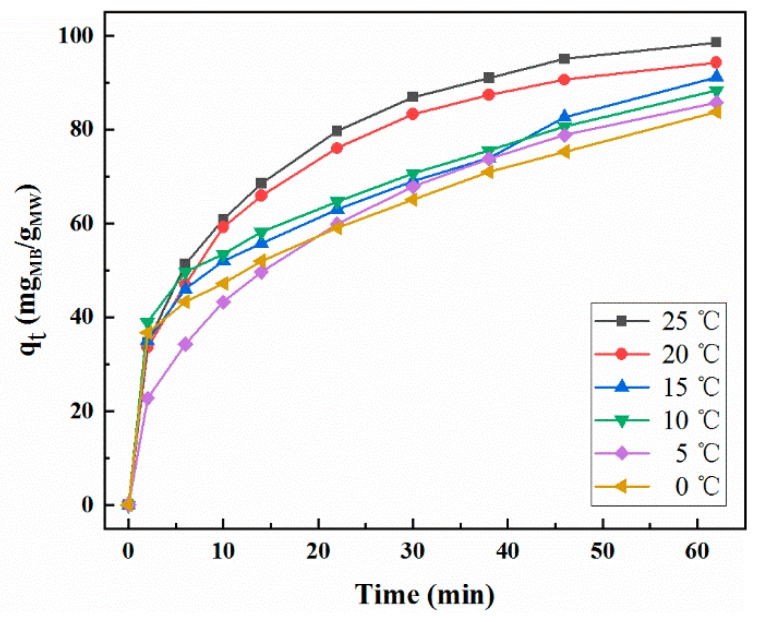
Relationship between adsorption capacity and time at different temperatures.

**Figure 6 molecules-25-00002-f006:**
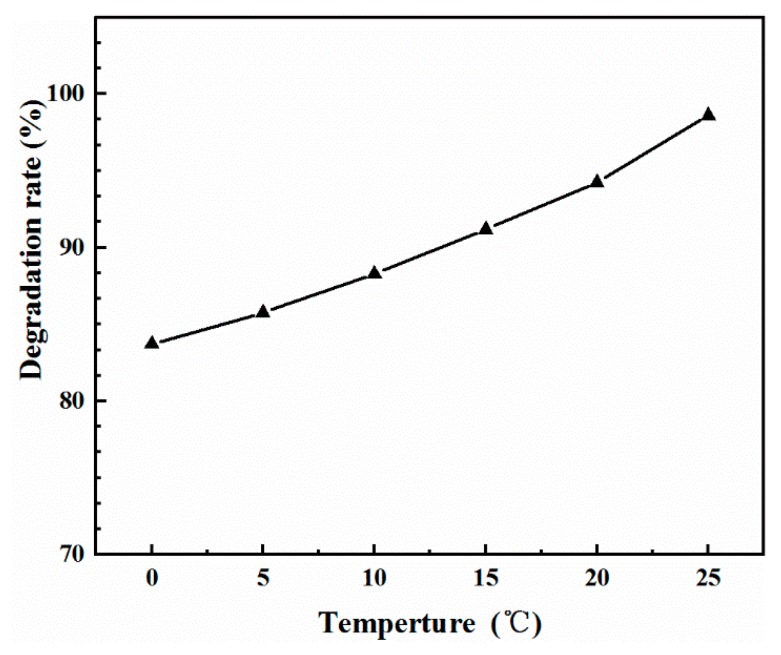
Effect of reaction temperature on the degradation rate of methylene blue (MB).

**Figure 7 molecules-25-00002-f007:**
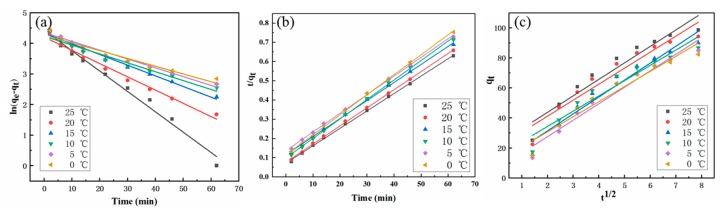
(**a**) pesudo-first order, (**b**) pesudo-second order, and (**c**) intra-particle diffusion model plots for adsorption data of MB onto MoS_2_/WO_3_ at different temperatures.

**Figure 8 molecules-25-00002-f008:**
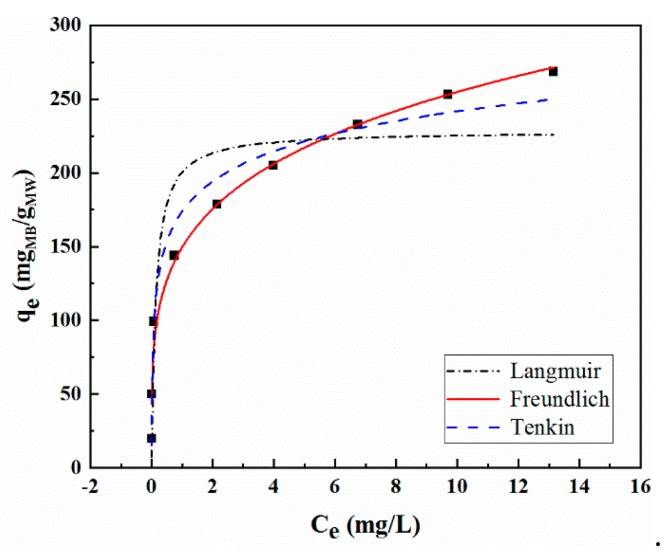
Fitting of equilibrium data to various adsorption isotherms.

**Table 1 molecules-25-00002-t001:** Kinetic parameters of the pseudo-first and pseudo-second order model and intra-particle diffusion model.

Temperature (°C)	q_e exp_ (mg/g)	Pseudo-First Order Model			Pseudo-Second Order Model			Intra-Particle Diffusion Model		
		k_1_ (min^−1^)	q_e cal_ (mg/g)	R^2^	k_2_·10^3^ (g·mg^−1^·min^−1^)	q_e cal_ (mg/g)	R^2^	k_p_ (g·g^−1^·min^1/2^)	C_i_	R^2^
25	99.599	0.0665	83.713	0.9853	1.1506	110.497	0.9992	10.9769	21.8190	0.9802
20	99.588	0.0433	67.078	0.9826	1.1562	106.383	0.9995	10.6596	19.8856	0.9806
15	99.559	0.0349	76.256	0.9858	0.7608	106.610	0.9992	11.2756	8.8740	0.9912
10	99.575	0.0290	69.030	0.9598	1.0374	98.814	0.9991	10.1768	14.0003	0.9851
5	99.565	0.0288	77.676	0.9772	0.6904	103.093	0.9992	10.9661	5.6302	0.9934
0	99.559	0.0254	71.621	0.9511	0.9474	96.061	0.9994	10.0647	10.6341	0.9397

**Table 2 molecules-25-00002-t002:** Summary of isotherm parameters.

Isotherms	Constants	MoS_2_/WO_3_
Langmuir	Q_m_	228.2849
	b	7.1529
	R^2^	0.8718
Freundlich	K_i_	161.5729
	1/m	0.2052
	R^2^	0.9898
Temkin	A	29.5634
	B	354.7873
	R^2^	0.9779

**Table 3 molecules-25-00002-t003:** The reported maximum monolayer coverage capacity of MB onto other adsorbents.

Adsorbents	Q_0_ (mg/g)	Reference
Flower-like sodium titanate	58.0	[43]
Jute fiber carbon	74	[44]
Perlite	94	[45]
Dehydrated wheat bran	122.0	[46]
MoS_2_	136.99	[47]
MoS_2_/WO_3_	268.42	This study

Q_0_ is the maximum adsorption capacity of a material reported in the literature under certain conditions.

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
