# Peer review of "The Combination of MoS2/WO3 and Its Adsorption Properties of Methylene Blue at Low Temperatures"

_molecules, 2019, doi:10.3390/molecules25010002_

Round 1
Reviewer 1 Report
Generally, this paper is well-written, well-organized and the data interpretation is clear. The only thing why MoS2/WO3 was utilized for adsorbent is still not clear. In terms of adsorption capacity, porous materials are better. For example, the Qmax value in Table 3 are not proper to compare adsorption capacity as porous silica are much better in adsorption (see review paper “European Journal of Environmental Sciences, Vol. 4, No. 2, pp. 116–130). So, I suggest authors to add-up explanation or data on those points.
1) The reason why adsorption property of MoS2/WO3 is important should be clarified. I agree that the catalytic effect of MoS2/WO3 is related to the adsorption. But, MB decomposition if not linearly correlated to the adsorption amount. Even with low adsorption capacity, catalytic effect can be achieved if the adsorbent-adsorbate interaction kinetic is brisk.
2) Authors claim that the chemical defect (interpreted by XPS) influenced the adsorption of MB. I fully agree that the XPS data clearly demonstrated the modified electronic structure and unsaturated bonding. However, it is not clearly that those unsaturated bonding had major effect in high MB adsorption. Rather specific surface area calculation is required for the MoS2/WO3 sample. If the unsaturated bonding is important in MB adsorption, authors should verify the “chemisorption” of MB on MoS2/WO3.
Author Response
Dear reviewer:
I am very greatful to your comments for the manuscript. The questions were answered in the attachment.

Reviewer 2 Report
The authors have investigated on the adsorption properties of the combination of MoS2/WO3 of MB at low temperatures.
The absorption of MB to remove it from water is an important problem. There are many contributions towards the solution to this problem (see some Refs. below).
On the other hand, I find some problems in this work. There are several points to be attended before considering it for publication.
-There are many typing mistakes.
-In the conclusions, the authors say: The XPS results show that besides MoS2 and WO3 existed in the products, there were also new chemical bonds formed. The transition metal oxide MoO3 and the sulfide WS2 both have good electro-active sites, so the ability of the composite material to adsorb positively charged dye is greatly improved.”
The authors do not mention how those new bonds were formed. What are the mechanisms to produce these reactions? They don´t mention the amount of MoO3 and WS2, nor the relative importance of these substances on the ability of the surface to adsorb MB.
-There is no mention of the previous investigations by other authors on activated carbon, graphene oxide, nor carbon nanotubes to adsorb MB. The last two systems are very similar to that considered by authors (see Refs 1-7 below).The efficiency of those methods is very similar to the one obtained in this report.
-The authors must show the advantages of their method over other methods to remove MB.
Some References:
-(AC) J. Hazard. Mater. 165, 1029–1039. 2009.
-(CNTs) J. Hazard. Mater. 196, 109–114.2011.
-(CNTS) Chem. Mater. 17, 3457–3463.2005.
-(GRAPHENE) Colloids Surf. A 90, 197–203. 2012.
-(GRAPHENE OXIDE) (GO) J. Colloid Interf. Sci. 359, 24–29. 2011.
-(ACTIVATED CARBON, GRAPHENE OXIDE, AND CARBON NANOTUBES) Chemical Engineering Research and Design 9 1 (2013) 361–368.
-Technol. 99, 1503–1508. 2008).
-Mater. 158, 531–540. 2008).
-J. Hazard. Mater. 185, 507–511. 2011.
-Chem. Eng. J. 168, 1193–1200. 2011).
-Dyes. Pigments 63, 243–250. 2004.
-Environmental Technology, 2019. DOI: 0.1080/09593330.2019.1585481.
Author Response

(The authors gave the same response as above.)

Round 2
Reviewer 1 Report
Authours represented ambiguous points of previous manuscrtipt. It is worth to be published.
Reviewer 2 Report
The authors have made important changes to their manuscript. It can be accepted for publication as it is.